# Carbonic Anhydrase VIII (CAVIII) Gene Mediated Colorectal Cancer Growth and Angiogenesis through Mediated miRNA 16-5p

**DOI:** 10.3390/biomedicines10051030

**Published:** 2022-04-29

**Authors:** Mingli Hsieh, Pei-Ju Huang, Pei-Yu Chou, Shih-Wei Wang, Hsi-Chi Lu, Wei-Wen Su, Yuan-Chiang Chung, Min-Huan Wu

**Affiliations:** 1Department of Life Science, Tunghai University, No. 1727, Sec. 4, Taiwan Boulevard, Taichung 407, Taiwan; mhsieh@thu.edu.tw; 2Life Science Research Center, Tunghai University, No. 1727, Sec. 4, Taiwan Boulevard, Taichung 407, Taiwan; peiyu67@thu.edu.tw (P.-Y.C.); hclu@thu.edu.tw (H.-C.L.); 3Department of Family Medicine, Changhua Christian Hospital, Changhua 500, Taiwan; triangle0128@gmail.com; 4Bachelor of Science in Senior Wellness and Sport Science, Tunghai University, No. 1727, Sec. 4, Taiwan Boulevard, Taichung 407, Taiwan; 5Senior Life and Innovation Technology Center, Tunghai University, No. 1727, Sec. 4, Taiwan Boulevard, Taichung 407, Taiwan; 6Graduate Institute of Natural Products, College of Pharmacy, Kaohsiung Medical University, Kaohsiung 807, Taiwan; shihwei@mmc.edu.tw; 7Food Science Department and Graduate Institute, Tunghai University, Taichung 407, Taiwan; 8Department of Gastroenterology and Hepatology, Changhua Christian Hospital, Changhua 500, Taiwan; mhwu@go.thu.edu.tw; 9Department of Surgery, Cheng-Ching General Hospital, Taichung 407, Taiwan; wingnice@gmail.com; 10Department of Surgery, Kuang Tien General Hospital, Taichung 407, Taiwan

**Keywords:** carbonic anhydrase VIII, vascular endothelial growth factor, angiogenesis, miR-16-5p

## Abstract

Carbonic anhydrase VIII (CAVIII) is a member of the CA family, while CA8 is the oncogene. Here we observed increased expression of CAVIII with high expression in colorectal cancer tissues. CAVIII is also expressed in more aggressive types of human colorectal cancer cells. Upregulated CAVIII expression in SW480 cell lines increased vascular endothelial growth factor (VEGF) and reduced miRNA16-5p. Conversely, knockdown of the CAVIII results in VEGF decline by up-regulated miRNA16-5p. Moreover, the collection of different grades of CAVIII expression CRC cells supernatant co-culture with endothelial progenitor cells (EPCs) promotes the ability of tube formation in soft agar and migration in the Transwell experiment, indicating that CAVIII might facilitate cancer-cell-released VEGF via the inhibition of miRNA16-5p signaling. Furthermore, in the xenograft tumor angiogenesis model, knockdown of CAVIII significantly reduced tumor growth and tumor-associated angiogenesis. Taken together, our results prove that the CAVIII/miR-16-5p signaling pathway might function as a metastasis suppressor in CRC. Targeting CAVIII/miR-16-5p may provide a strategy for blocking its metastasis.

## 1. Introduction

Colorectal cancer (CRC) is the third most common malignant tumor and the fourth leading cause of cancer death in the world. It has also been ranked in the top three for cancer incidence and mortality rate in Taiwan’s general population since 1982 [1,2]. The incidence and mortality of CRC continue to increase due to metastatic disease and poor treatment outcomes, necessitating the need to better understand the underlying mechanisms that lead to tumorigenesis and progression. Despite significant advances in the treatment of CRC in recent years, the survival rate of metastatic colorectal cancer (mCRC) remains low [3,4].

As effective treatment options for mCRC remain limited, there remains an urgent clinical need for new treatment strategies. Therefore, we identified protein variants of human carbonic anhydrase VIII (CAVIII) in CRC patients through integrated bioinformatics analysis of The Cancer Genome Atlas Program (TCGA) and The Human Protein Atlas (HPA) databases. CA is a zinc-containing metalloenzyme that catalyzes the synthesis of carbonic acid from carbon dioxide and water. CAVIII, which belongs to the a- family, lacks a binding site for zinc ions, resulting in a loss of the ability to catalyze carbon dioxide hydration, but CAVIII gene products have important functions [5]. CAVIII can compete with IP3 (Inositol triphosphate) for the coupling site of IP3R1 (Inositol trisphosphate receptor), which regulates intracellular calcium ion release. This is related to neuronal excitability, an outgrowth of nerve processes, the release of neurotransmitters, mitochondrial energy production, and cell fate (neuronal excitability, outgrowth of nerve processes, release of neurotransmitters, mitochondrial energy production, and cell fate). The effects of CA on neuronal excitability, the outgrowth of nerve processes, the release of neurotransmitters, mitochondrial energy production, and cell fate are considerable. In early studies, CAVIII was found to be associated with neurodegenerative diseases, including epilepsy, MERRF disease, and spinal cerebellar ataxia (SCA) [6,7]. 

However, it has been shown that CAVIII is an oncofetal antigen that plays a role in the oncogenesis of non-small cell lung and colon cancers [8,9]. Nishikata et al. found that excessive expression of CAVIII in human LOVO colorectal cancer cells resulted in increased proliferation and invasion of cancer cells [10]. A recent study in renal carcinoma (RCC) cells found that CAVIII was overexpressed in Caki-1 and 769-P cells and increased the expression of pAKT and MMP2, thus promoting the proliferative and migratory ability of cancer cells [11]. In conclusion, it is suggested that although CAVIII can promote the proliferative and migratory ability of CRC cells, further mechanistic studies are needed to investigate the mechanism through which CAVIII is involved in CRC.

However, the formation and progression of CRC are inextricably linked to angiogenesis, which plays an important role in the proliferation and metastasis of CRC [12]. With the continuous research on anti-angiogenic therapies over the years, targeting angiogenesis has become an important therapeutic strategy for many tumors, including CRC; for example, the proliferation, apoptosis, migration, invasion, and angiogenesis of colorectal cancer [13]. Previous studies have indicated that signaling pathways associated with Vascular endothelial growth factor (VEGF-A) are regulated by miRNAs, such as miR-29, miR-16, and the miR-27 family [14,15,16,17]. Among them, mR-16-5p is also the first confirmed cancer-related gene in miRNAs [18] and has been reported to have an abnormal expression in the tumor tissues and blood of tumor patients, as well as being downregulated in most cancer cell lines. Aberrant expression of miR-16-5p promotes tumor cell proliferation, metastasis, and angiogenesis, and can also affect the treatment sensitivity, such as radiotherapy and chemotherapy [19,20,21]. However, the details of the mechanisms by which these miRNAs affect VEGF are unclear, and the role of CAVIII in tumor angiogenesis is largely unknown. Therefore, in this study, we examined the relationship between CAVIII and VEGF-A expression and tumor angiogenesis and further investigated the molecular mechanism of CAVIII-induced VEGF-A-dependent angiogenesis in the CRC microenvironment.

## 2. Materials and Methods

### 2.1. Chemicals and Reagents

We purchased anti-rabbit and anti-mouse IgG-conjugated horseradish peroxidase (Santa Cruz Biotechnology, CA, USA), mouse monoclonal antibodies specific for VEGF-A (Abcam, Cambridge, MA, USA), as well as rabbit polyclonal antibodies specific for CAVIII, MMP2, and MMP9 (Cell Signaling Technology, Danvers, MA, USA). The miR-16-5p, miRNA control, Lipofectamine 2000, and Trizol were purchased from Life Technologies (Carlsbad, CA, USA). ON-TARGETplus shRNAs of CA8 (product ID: V2LHS-_172768) were purchased from Dharmacon Research (Lafayette, CO, USA). Dulbecco’s modified Eagle’s medium (DMEM), fetal bovine serum (FBS), and all other cell culture reagents were from Gibco-BRL Life Technologies (Grand Island, NY, USA). Endothelial cell growth supplements (ECGS) were obtained from Millipore (Billerica, MA, USA). The BD Matrigel Matrix (#356,237) was purchased from Corning (Bedford, MA, USA). All other chemicals were purchased from Sigma-Aldrich (St. Louis, MO, USA).

### 2.2. Cell Culture

The human colorectal cell line (SW480; adenocarcinoma; Dukes’ type B cell line separated from 50-year-old Caucasian male) was purchased from the Bioresource Collection and Research Center (BCRC) (Hsinchu, Taiwan). SW480 was maintained in 90% Leibovitz’s L-15 medium with 2 mM L-glutamine, penicillin (100 U/mL), streptomycin (100 μg/mL), and 10% FBS at 37 °C with 5% CO_2_. The Endothelial progenitor cells (EPCs) culture program was approved by the Institutional Review Board of Mackay College of Medicine, New Taipei City, Taiwan (reference number P1000002). EPCs were cultured in an MV2 complete medium that contained the MV2 basal medium and growth supplement (PromoCell, Heidelberg, Germany) and was supplemented with 20% defined FBS (HyClone, Logan, UT, USA). Cultures were seeded on 1% gelatin-coated plasticware and maintained at 37 °C in a humidified 5% CO_2_ atmosphere. The preparation of EPC has been described before [22,23].

### 2.3. Construction of Stable Expression CA8 shRNA Cell Line

The CA8 shRNA was purchased from the National RNAi Core Facility (NRC) (Academia Sinica, Taipei, Taiwan). Full-length CA8 cDNA was amplified from pcDNA3.1-CA8-myc-His (Chang, unpublished work) using the synthesized primers CA8-NheI (5′-TCCTGATGCTAATGGGGAATACCAG-3′) and CA8-PmeI (5′-CTAAGAGGCTGAGTGGGCCGAAAG-3′). After cells were seeded at 24 h, the growth media was removed and replaced with fresh media containing polybrene (5 μg/mL). Cells were infected by adding the CA8 shRNA lentiviral particles to the culture and incubating them overnight. A lentiviral transfer vector containing the full-length CA8 with myc tagged was generated by PCR amplifying CA8-myc DNA fragment and inserted between NheI and EcoRI of pLKOAS3w.puro. The lentiviral short hairpin RNA (shRNA) expression vectors of SW480 shluc (pLKO.1-shLuc) and SW480 shCA8 (TRCN000153276) were used. The media was removed and replaced with 2 μg/mL puromycin to select stable transfectants. Single clones were picked, and the ectopic expression of the gene of interest was verified using Western blot analysis [24].

### 2.4. Transient Transfection

SW480 cells were cultured in a 6-well plate, and the miR-16-5p mimic was transfected into the cells by Lipofectamine™ 2000. ON-TARGETplus siRNAs (100 nM) were transiently transfected with the DharmaFECT1 transfection reagent, according to the manufacturer’s instructions [25].

### 2.5. Conditioned Medium (CM) Preparation

SW480 shluc cells and SW480 shCA8 cells were incubated. The cells were washed and transferred to a serum-free medium. The conditioned medium (CM) was then collected 2 days after the change of medium and stored at −80 °C until use.

### 2.6. ELISA Assay

SW480 shluc cells and SW480 shCA8 cells were cultured in 6-well plates. After reaching confluence, cells were changed to a serum-free medium. The conditioned medium (CM) was collected, the medium was removed, and was stored at –80 °C. Then, VEGF-A in the medium was determined using the VEGF-A ELISA kit (PeproTech, Rocky Hill, NJ, USA) according to the manufacturer’s protocol.

### 2.7. Migration Assay

The EPC migration assay was performed using an 8-mm pore size Transwell chambers (Coring, Coring, NY, USA). EPCs (1 × 10^4^ cells/well) were seeded onto the top chamber with the MV2 complete medium and then incubated in the bottom chamber with 50% MV2 complete medium and 50% colorectal cell CM.

### 2.8. Tube Formation Assay

Matrigel (BD Biosciences, Bedford, MA, USA) was dissolved at 4 °C overnight, and 48-well plates were prepared with 150 μL Matrigel in each well and then incubated at 37 °C for 30 min. After gel formation, EPCs (1 × 10^4^ cells) were seeded per well on the layer of polymerized Matrigel in cultured media containing 50% MV2 complete medium and 50% osteosarcoma cell CM, followed by incubation for 16 h at 37 °C. Then, the EPC tube formation was taken with the inverted phase-contrast microscope. Tube branches and total tube length were calculated using MacBiophotonics Image J software [26].

### 2.9. Immunohistochemical (IHC) Staining

Human colorectal tissue was examined at Changhua Christian Hospital and Cheng Ching General Hospital, and all experimental procedures were approved by the Institutional Review Board (IRB No. 160316 and No. 150025). Tissue microarrays contained 100 cases of formalin-fixed and paraffin-embedded CRC tissues samples that underwent curative surgery. No patient received preoperative chemotherapy or radiotherapy. Clinical and pathological reports were reviewed for age, sex, differentiation, clinical stage, pathological tumor (pT) stage, lymph node metastasis, distant metastasis, radiotherapy, chemotherapy, and recurrence. The tumors were recruited by the stage level of CRC, with 3 patients in stage 1, 4 patients in stage 2, 12 patients in stage 3, and 4 patients in stage 4. Human colorectal tissue was purchased, and sections were deparaffinized with xylene and rehydrated by adding ethanol. Endogenous peroxidase activity was blocked with 3% hydrogen peroxide in methanol for 10 min. Heat-induced antigen retrieval was carried out for all sections in 0.01 M sodium citrate buffer at pH 6 at 95 °C for 25 min. Human CAVIII and VEGF-A antibodies were applied at a dilution of 1:300 and incubated at 4 °C overnight. The antibody-binding signal was detected using the NovoLink Polymer Detection System (Leica Microsystems, Wetzlar, Germany) and visualized with the diaminobenzidine reaction. The sections were counterstained with hematoxylin. The immunohistochemistry results were scored by taking into account the percentage of positive detection and intensity of the staining.

### 2.10. Western Blot Analysis

The protein concentration was determined using the Thermo Scientific Pierce BCA Protein Assay Kit (Thermo Fisher Scientific Inc., Waltham, MA, USA). Proteins were resolved on SDS-PAGE and transferred to immobilon polyvinyl difluoride (PVDF) membranes. The blots were blocked with 4% BSA for 1 h at room temperature and incubated with primary antibodies for 1 h at room temperature. After three washes in Tris-buffered saline with 0.05% Tween 20 (TBS-Tween), the blots were subsequently incubated with a donkey anti-rabbit or anti-mouse peroxidase-conjugated secondary antibody for 1 h at room temperature. The blots were visualized by enhanced chemiluminescence using Kodak X-OMAT LS film (Eastman Kodak, Rochester, NY, USA). Quantitative data were obtained using a computing densitometer and ImageQuant software (Molecular Dynamics, Sunnyvale, CA, USA).

### 2.11. Quantitative Real-Time PCR

Total RNA was extracted from SW480 shluc cells and SW480 shCA8 cells using a TRIzol kit (MDBio Inc., Taipei, Taiwan). The reverse transcription reaction was performed using 2 μg of total RNA that was reverse transcribed into cDNA using an oligo (dT) primer. Quantitative real-time polymerase chain reaction (q-PCR) analysis was carried out using TaqMan^®^ one-step PCR Master Mix (Applied Biosystems, Foster City, CA, USA). Total complementary DNA (100 ng/25 μL reaction) was mixed with sequence-specific primers and TaqMan^®^ probes according to the manufacturer’s instructions. Sequences for all target gene primers and probes were purchased commercially (β-actin was used as the internal control) (Applied Biosystems). Q-PCR assays were carried out in triplicate using a StepOnePlus sequence detection system. The cycling conditions were 10 min of polymerase activation at 95 °C, followed by 40 cycles at 95 °C for 15 s and 60 °C for 60 s.

### 2.12. In Vivo Tumor Angiogenesis Assay

SW480 shluc cells and SW480 shCA8 cells were then collected. Ten male BALB/c nude mice (4 weeks of age; purchased from the National Laboratory Animal Center, Taipei, Taiwan) were used and randomized into two groups: SW480/control Luc and SW480/CA8 shRNA. Each group was subcutaneously injected with 0.4 mL SW480 shluc cells and SW480 shCA8 cells. After 10 days, the mice were killed by overdose with an anesthetic. The tumor was removed and fixed in 10% formalin. The tumor volume, weight, and hemoglobin concentration of the tumor were measured.

### 2.13. Mice Xenograft Assay

Male BALB/c nude mice (5 weeks old) were randomly divided into 2 groups (5 mice per group). SW480 shluc cells and SW480 shCA8 cells (2 × 10^6^ cells per mouse) were resuspended in a serum-free medium with Matrigel at a 1:1 ratio, and then subcutaneously injected into the right flank of each animal. Mice body weights were recorded twice weekly. Tumor volume was monitored by the Xenogen IVIS system and images were captured 10 min after D-luciferin injection with a 60-s exposure using a CCD camera. After 21 days, mice were euthanized by subjecting them to CO_2_ inhalation and the tumor volume was calculated using the formula V = (LW2)π/6, where V is the volume (mm^3^), L is the largest diameter (mm), and W is the smallest diameter (mm).

### 2.14. Hemoglobin Assay

All the Matrigel plugs and tumors were processed to measure blood vessel formation. Briefly, the concentration level of hemoglobin in the vessels that invaded the Matrigel was determined with Drabkin’s reagent (Sigma-Aldrich) according to the manufacturer’s instructions. Taking the same weight of plugs or tumors, they were homogenized in 1 mL of RIPA lysis buffer and then centrifuged at 1000 rpm. Then, 20 μL of supernatants were added to 100 μL of Darkin’s solution. The mixture was allowed to stand for 30 min at room temperature, and then readings were taken at 540 nm in a spectrophotometer for the in vivo xenograft model and tumorigenicity.

### 2.15. Statistical Analysis

Results are presented as means ± standard deviations (SD). The statistical analysis was executed with Predictive Analytics Suite Workstation (PASW) Statistical software for Windows, Version 18.0 (SPSS Inc., Chicago, IL, USA) using a one-way ANOVA followed by Scheffe’s test. A value of *p* < 0.05 was considered statistically significant.

## 3. Results

### 3.1. Identification of CAVIII and VEGF-A Expression in Colorectal Cancer Tissues

To confirm the expression of CAVIII and VEGF-A, we collected and tested tissues from 20 patients with different grades of colorectal cancer at Cheng Ching Hospital. Tissue samples were then stained for CAVIII and VEGF-A in immunohistochemistry (IHC) assays. Subsequently, the expression of CAVIII and VEGF-A was analyzed in all samples and the mean optical density was calculated using ImageJ software. The results showed that the expression of CAVIII and VEGF-A proteins in colorectal cancer clinical stage tumor tissues increased with the severity of the stage and were significantly correlated with each other (Figure 1A–D), Appendix A.

### 3.2. Knockdown of CAVIII or Over-Expression of CAVIII Directly Regulates VEGF-A Expression in Human Colon Cancer Cell Lines

To date, there is no detailed information on the relationship between CAVIII and VEGF-A in human colon cancer cells. Here, we next infected SW480 cells with lentivirus-expressing shRNA targeting CAVIII (infected SW480 cells with lentivirus-expressing shRNA targeting CA8) and established CAV8 overexpression cell lines (control EGFP/myc and CA8 overexpression/myc) to observe the effect on CAVIII (Figure 2A–C). We also observed the effect of VEGF-A expression on these different types of SW480 cell lines. The results showed that knockdown of CAVIII decreased the expression of the VEGF-A mRNA and VEGF-A protein and the excretion of the VEGF-A protein by SW480, and conversely, once CA8 was over-expressed, the mRNA and supernatant protein expression of VEGFA are significantly elevated (Figure 2D–F).

### 3.3. Knockdown of CA8 Over-Expression of CAVIII Directly Regulates VEGF-A Expression through Regulated by miR16-5p

The intricate networks of miRNAs in CAVIII-induced angiogenesis are little understood. Our results indicate that CAVIII promotes VEGF-A expression. Therefore, we undertook a bioinformatics analysis using open-source database software (TargetScan: http://www.targetscan.org/vert_72/, accessed on 4 April 2022; miRWalk: http://zmf.umm.uni-heidelberg.de/apps/zmf/mirwalk2/, accessed on 4 April 2022 and miRBase: https://www.mirbase.org/, accessed on 4 April 2022) to screen for miRNAs that regulate VEGF-A expression (Figure 3A). To validate these findings, we compared the expression levels of miR16-5p in SW480 under graded CAVIII expression. We found that CA8 overexpression suppressed the extent of miR16-5p expression. In addition, the miR16-5p increased in CA8 knockdown expression SW480 cell lines (Figure 3A right). Similarly, the mRNA and protein level expression of VEGF-A is also regulated by CAVIII expression and directly mediated by miR16-5p. The expression of VEGF-A was inhibited in CA8 shRNA CRC and reversed by treatment with the miR16-5p inhibitor (Figure 3B,C). Conversely, CA8 overexpression CRC cells increased VEGF-A amounts and were reduced by the miR16-5p mimic (Figure 3D,E). These results suggest that CAVIII affects the expression of VEGF-A by regulating miR16-5p.

### 3.4. CAVIII Influences VEGFA Production and Affects In Vitro Angiogenesis by Regulating miR-16-5p 

We further verified that the expression of CAVIII directly affects angiogenesis, and we designed in vitro angiogenesis assays through endothelial progenitor cells (EPCs). New vessel formation also involves EPCs, and since the formation of new blood vessels depends on the migration and tube formation of EPCs through the basement membrane of capillaries, we first analyzed the role of CAVIII in the migration activity of EPCs. In this experiment, we collected different levels of CAVIII cell lines and supernatants administered with the miR16-5p mimic or inhibitor for analysis with EPCs and established a TransWell assay to simulate microangiogenesis in vitro (Figure 4A). As the results showed, CA8 overexpression CRC cell lines increase the migration ability of EPCs after incubation with a conditioned medium (CM), and conversely, EPC migration was significantly reduced when pretreating with the miR16-5p mimic (Figure 4C). On the other hand, the migratory ability was inhibited via incubation with the CA8 shRNA CRC CM. Furthermore, it was reversed via pretreatment with the miR16-5p inhibitor (Figure 4B). Meanwhile, the capillary formation assays also showed that the knockdown of CA8-treated SW480 cells was able to reduce capillary network structure formation and reorganization (Figure 5A). Once miR16-5p inhibitors are administered externally, the expression of angiogenesis is elevated. Similarly, when SW480 expressed CAVIII in large amounts, it increased the amount of angiogenesis but was also inhibited by miR16-5p (Figure 5B,C). These shreds of evidence clearly demonstrate that CAVIII expression directly regulates the angiogenic effects of CRC through the modulation of miR16-5p.

### 3.5. Inhibition of Tumor Growth and Angiogenesis by CAVIII Expression in a Mouse Xenograft Model 

We demonstrated CAVIII-mediated promotion of tumor growth in an in vivo mouse xenograft model, as shown in Figure 6A–D, and showed that knockdown of CAVIII inhibited tumor growth in mice. We also measured the concentration of hemoglobin in tumor specimens and showed that knockdown of CA reduced the concentration of hemoglobin in tumors (Figure 6E,F). Immunohistochemistry (IHC), RT-qPCR, and Western blot results of tumor specimens also showed that the expression of CAVIII and VEGF-A was reduced after Knockdown of CAVIII (Figure 6G,H). Overall, these results suggest that the expression of CAVIII can affect the angiogenesis of tumors in vivo.

## 4. Discussion

Tumor angiogenesis is dependent on a highly complex program of growth factor signaling (i.e., VEGF, transforming growth factor (TGF), alkaline fibroblast growth factor (bFGF), and platelet-derived growth factor (PDGF)), endothelial cell (EC) proliferation, extracellular matrix (ECM) remodeling, and stromal cell interactions. The importance of pro-angiogenic inducers in tumor growth, invasion, and metastasis has made them excellent therapeutic targets for many cancers [22,23,27,28,29]. Over the past decade, the number of anti-angiogenic drugs developed for cancer therapy through targeting has increased, with at least 80 drugs being studied in preclinical studies and phase I-III clinical trials, yet the number of drugs that are clinically effective is still very limited and requires additional research investment [30,31,32]. In this study, we found a high expression of CAVIII in human colorectal cancer tissues and a significant increase in VEGFA, a key protein for vascular neovascularization. Furthermore, if knockdown of CAVIII decreased VEGFA expression in cellular assays, it also decreased in vitro vascularization (TransWells & Tubeformation), and conversely, VEGFA and in vitro vascularization were significantly increased in CRC cell lines with CAV8 overexpression. Our study further investigated the effect of CAVIII on miRNAs and found that CAVIII increased VEGF-A performance by downregulating miR16-5p expression and VEGFA production and angiogenesis of EPCs after external administration of the miR16-5p mimic. Furthermore, in a mouse xenograft model, cell lines with reduced CAVIII expression reduced tumor growth, hemoglobin concentration, and angiogenesis, suggesting that CAVIII/miR16-5/VEGFA may be a new target for the treatment of colorectal cancer.

Previous studies have shown that the family can play a key role in tumorigenesis and growth development, especially CA9, 10, and 12 [11,33,34]. In studies of lung cancer and osteosarcoma, CA8 is considered an oncogene. Moreover, in renal cell carcinoma (RCC) tumor, knockdown of CAVIII is followed by the significant suppression of cell proliferation and invasive migration in RCC (Caki-1 and 769-P) [35,36]. Our previous studies found that overexpression of CAVIII in human OS (HOS) cells significantly increased cell proliferation both in vitro and in vivo. In contrast, the downregulation of CAVIII in HOS cells reduced cell invasion and colony formation ability in soft agar, and these results are consistent with the present study [24]. Nevertheless, there are still few studies on the effect of CAVIII on cancer. This is the first study on the effect of CAVIII on CRC angiogenesis, and thus provides critical evidence not only of the potential significance of CA8 as a CRC oncogene, but also direct evidence of the effect of CAVIII on CRC angiogenesis. This indicates that the mechanism of action of CAVIII is not only of traditional neurological importance but also has the potential to have a critical impact on the tumor microenvironment.

MicroRNAs are small, single-stranded, non-coding RNA molecules involved in the post-transcriptional regulation of various genes, the dysregulation of which can lead to tumorigenesis. One of the genes regulated by miRNAs is VEGF-A, which is responsible for angiogenesis. In this study, miRDB and miRwalk were used to identify the corresponding miRNAs at the VEGF-A gene location, and miR-16-5p was found to be the most influential direct target in the CRC cell line with CA8 gene inhibition overperformance. Some past studies on cancer angiogenesis, in breast cancer cells and nude mice, also found miR-p16-5p to be a possible target of VEGFA. Similarly, in chondrosarcoma studies, resistin and VEGF-A expression were positively correlated, resistin and VEGF-A were negatively correlated with miR-16-5p, and resistin inhibited miR-16-5p expression through PI3K/Akt signaling to promote VEGF-A expression and angiogenesis [37,38].

Here we found that the amount of CAVIII expression directly affects VEGF-A mRNA, VEGF-A protein expression, and angiogenesis. Once knockdown of CAVIII downregulates miR16-5p expression, the combination of miR-16-5p with CAVIII and VEGFA-related signaling pathways may be an effective molecule for future colorectal cancer patients. Drug resistance in CRC is often the main cause of treatment failure, and clinical treatment with anti-angiogenic drugs as adjuvant therapy can effectively improve the therapeutic effect. Thus, these findings highlight the potential therapeutic value of miR-16-5p in colorectal cancer and provide potential new drug therapies with the ultimate goal of overcoming drug resistance and improving clinical outcomes in cancer patients.

## Figures and Tables

**Figure 1 biomedicines-10-01030-f001:**
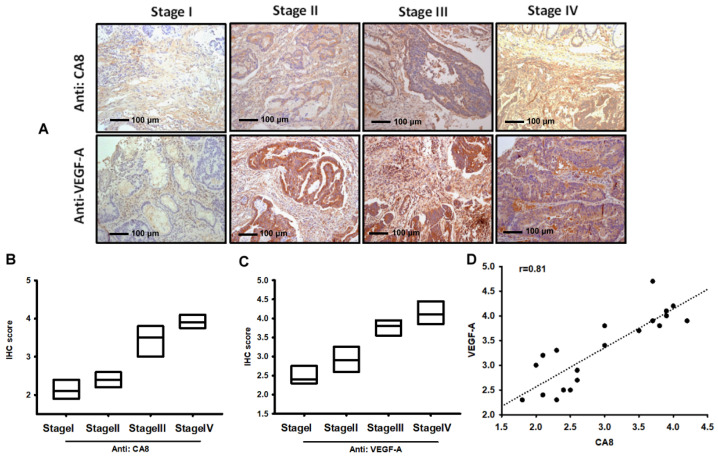
The correlation of CA-8, VEGF-A, and tumor grades in human colorectal cancer (CRC) tissues. IHC of CAVIII and VEGF-A (**A**–**C**) expressions in Stage I (*n* = 3), Stage II (*n* = 4), Stage III (*n* = 6), and Stage IV (*n* = 4) CRC tissues. The correlation and quantitative data are shown in (**D**).

**Figure 2 biomedicines-10-01030-f002:**
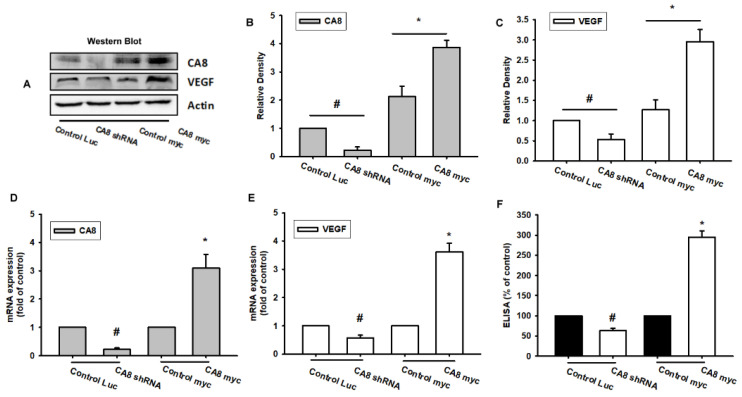
CAVIII reduces VEGF expression in knockdown CA8 CRC cells and increases VEGF expression in CA8 overexpression CRC cells. CRC cell lines included SW480/control Luc, SW480/CA8 shRNA (CA8 knockdown expression), SW480/Control myc, and SW480 CA8 myc (CA8 overexpression). (**A**–**F**) Comparison of the CAVIII or VEGF expression in different grades of CAVIII expression in CRC cells was examined by qPCR, ELISA, and Western blotting. Results are expressed as the mean ± SEM. * *p* < 0.05 compared with controls. # *p* < 0.05 compared with the control group.

**Figure 3 biomedicines-10-01030-f003:**
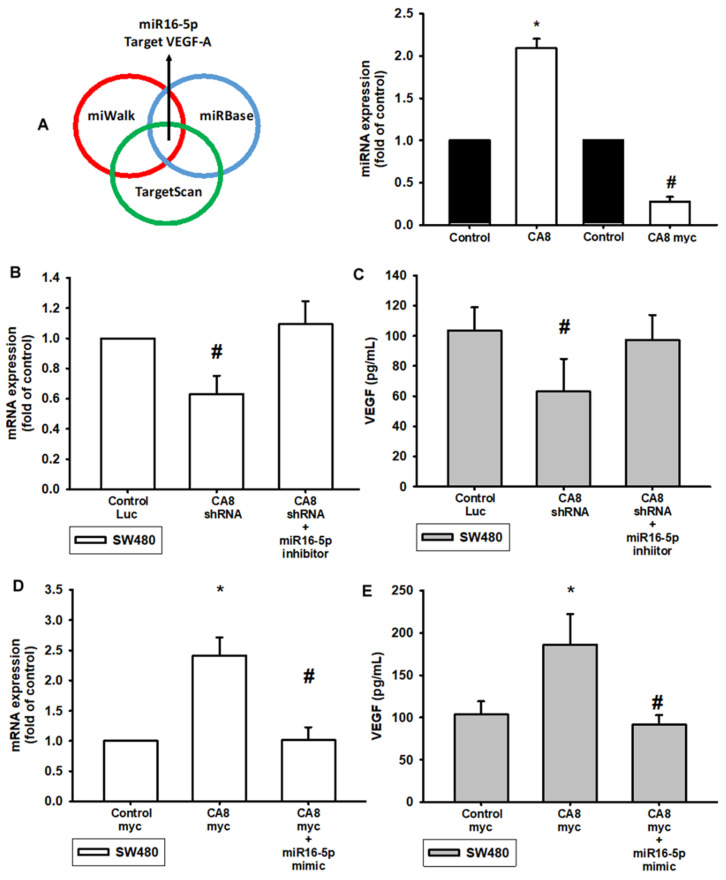
CAVIII increases production VEGF-A by suppressing miR16-5p expression. (**A**) Open-source software (TargetScan, miRDB, and miRWalk) sought to identify miRNAs that could possibly interfere with VEGF-A. SW480 cells were transfected with the indicated plasmids (CA8 shRNA or CA8 myc, 400 ng per well) and RNA was collected and extracted 24 h later. (**B**,**C**) VEGF mRNA and protein expression in CA8 knockdown cell lines (CA8 shRNA) and treatment with miR-16-5p inhibitor examined by qPCR and Western blot. (**D**,**E**) VEGF mRNA and protein expression in CA8 overexpression cell lines (CA8 myc) and treatment with miR-16-5p mimic examined by qPCR and Western blot. Results are expressed as the mean ± SEM. * *p* < 0.05 compared with controls; # *p* < 0.05 compared with the CA8 myc group.

**Figure 4 biomedicines-10-01030-f004:**
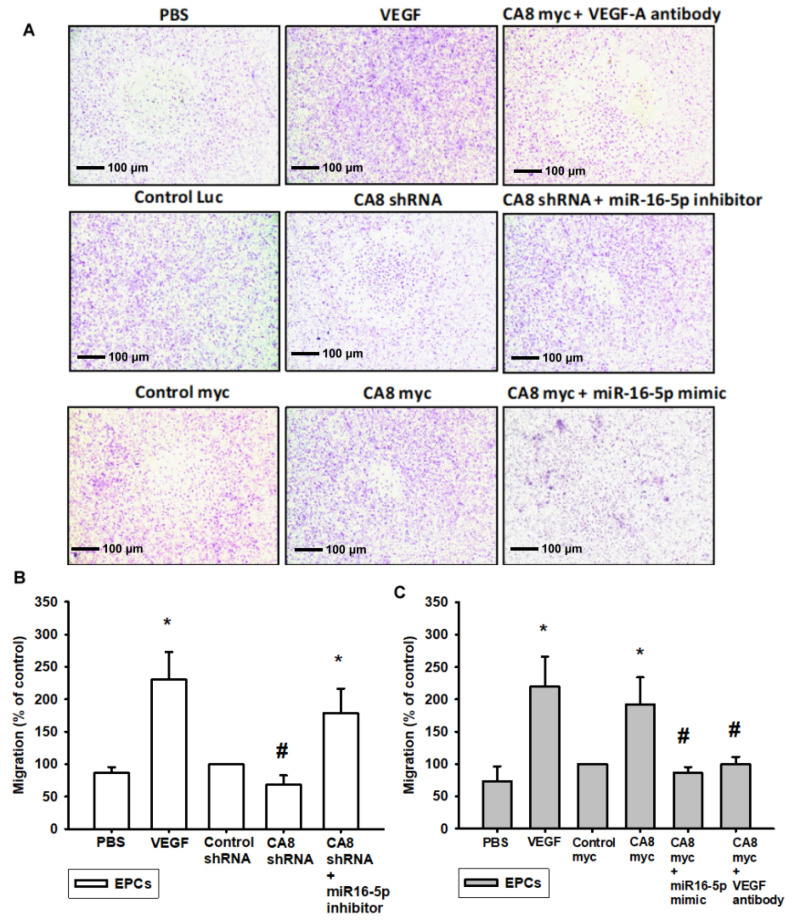
Effects of CAVIII on VEGF-A-induced migration in human EPCs. EPCs were incubated with various grades of CAVIII expression CRC cells for 24 h. CRC Cells were transfected with miR16-5p mimic or inhibitor then the conditioned medium (CM) was then collected and applied to endothelial progenitor cells (EPCs). (**A**–**C**) Cell migration was examined by Transwell assays, respectively. Data represent the mean ± S.E.M. of four independent experiments. * *p* < 0.05 compared with the control group; # *p* < 0.05 compared with the CA8 myc group. Original magnification 200×. Scale bar; 100 μm.

**Figure 5 biomedicines-10-01030-f005:**
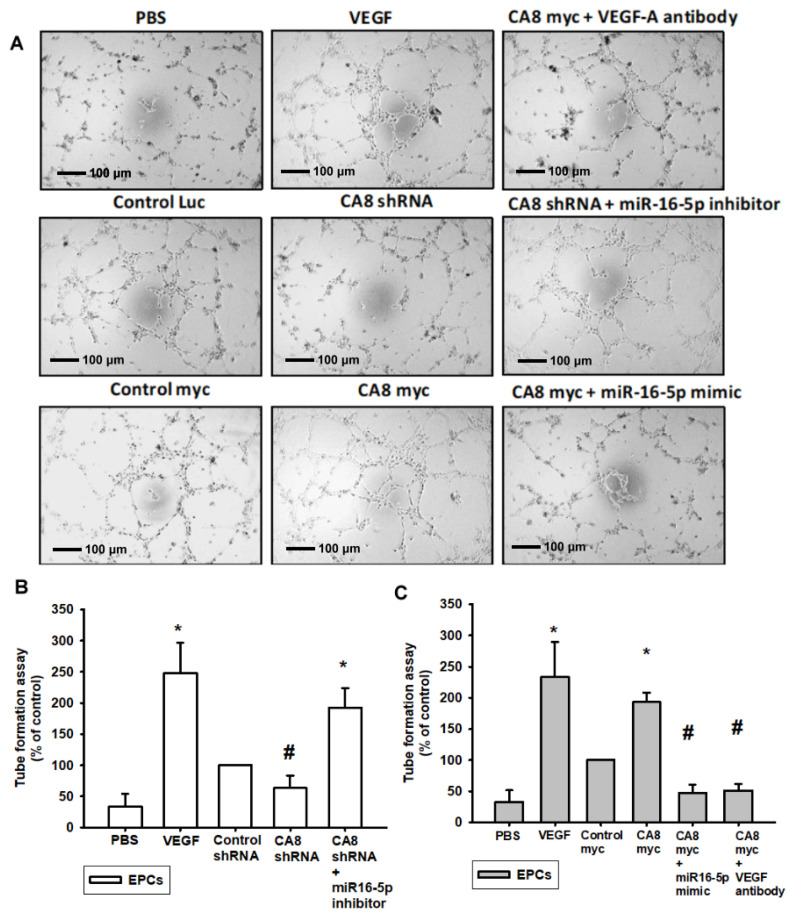
Effects of CAVIII on VEGF-A-induced capillary-like structure formation in human EPCs. EPCs were incubated with various grades of CAVIII expression CRC cells for 24 h. CRC. Cells were transfected with miR16-5p mimic or inhibitor then the conditioned medium (CM) was then collected and applied to endothelial progenitor cells (EPCs). (**A**–**C**) Cell capillary-like structure formation was examined by tube formation assays, respectively. Data represent the mean ± S.E.M. of four independent experiments. * *p* < 0.05 compared with the control group; # *p* < 0.05 compared with the CA8 myc group. Original magnification 200×. Scale bar; 100 μm.

**Figure 6 biomedicines-10-01030-f006:**
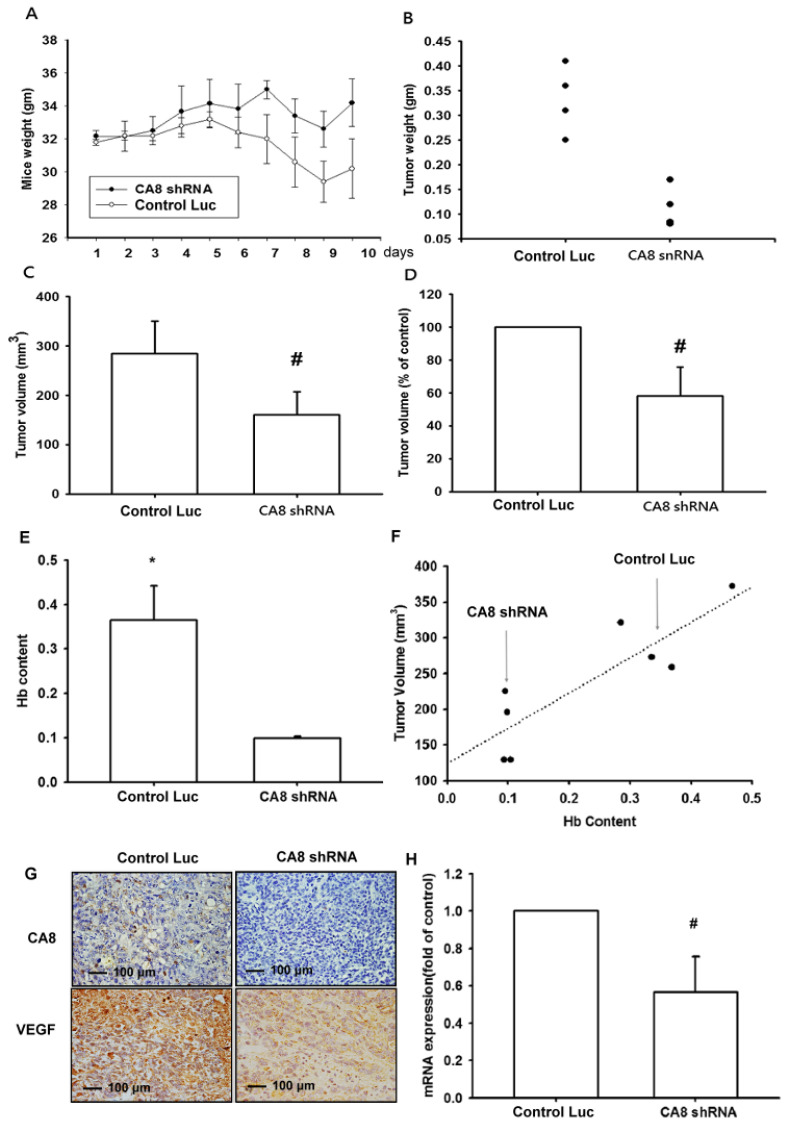
Knockdown of CAVIII decreases tumor-associated angiogenesis in mice. (**A**–**E**) SW480/control luc or Sw480/CA8 shRNA cells were mixed with Matrigel and injected into flank sites of mice for 10 days, and then resected. The tumors were photographed with a microscope, weight and volume were measured, and the hemoglobin levels were quantified. The correlation between tumor volume and hemoglobin levels is shown in (**F**). (**G**,**H**) The tumor was excised from mice and photographed, stained with CAVIII or VEGF-A, and quantified the mRNA content. Results are expressed as mean ± s.e. * *p* < 0.05 compared with CA8 shRNA, # *p* < 0.05 as compared with the control group. Original magnification 200×. Scale bar; 100 μm.

## Data Availability

Not applicable.

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
