# Peer review of "Carbonic Anhydrase VIII (CAVIII) Gene Mediated Colorectal Cancer Growth and Angiogenesis through Mediated miRNA 16-5p"

_biomedicines, 2022, doi:10.3390/biomedicines10051030_

Round 1

Reviewer 1 Report

The ms by Hsieh et al. investigated over the role of CA8/miR‐16-5p signaling pathway in tumor development. The findings of the work are of relevance in the field, paving the way to novel targetable pahtways for tumor treatment. However, the authors should distinghuish between the protein, that is CA VIII and the gene that is CA8. Accordingly, the whole text has to be revised. CA VIII should be mostly used, such as in the first sentence of the abstract: CA VIII is the member of the CA family, while CA8 is the oncogene.

The ms can be thus published in the journal upon these minor revision.

Line 22 Rephrase these two sentences “Carbonic anhydrase 8 is a member carbonic anhydrase family member, also an oncogene. Here we observed increased expression of CA8 is high expression in colorectal cancer tissues. CA8 is also expressed in more aggressive types of human colorectal cancer cells.”

Line 331 correct “tumo”

Line 331 “exhavetosis”?

Reviewer 2 Report

In the manuscript „Carbonic Anhydrase VIII (CA8) Mediated Colorectal Cancer Growth and Angiogenesis Through Mediated miRNA 16-5p“  Mingli Hsieh et al. present an interesting study about the role of CA VIII in colorectal cancer  and angiogenesis. While this study is of scientific interest, the manuscript requires a major revision and conclusions of this publication are not supported by the presented results.

The main shortcoming of the work is that the authors present the results on only one type of CRC cells - SW480.

The results lack evidence of CA VIII, VEGF-A expression in the parental line SW480. Fill in this information.

Despite the fact that the role of miRNA 16-5p in the process of angiogenesis in CRC is described in the work, in the Introduction section the authors do not devote sufficient space to miRNA 16-5p. It is therefore necessary to supplement this information.

In the Materials and methods section, it is necessary to add data on the dilutions of the antibodies used.

In section 2.12 In vivo tumor angiogenesis assay, the authors describe the use of CM SW480 shluc and SW480 shCA8. However, mice were further injected with 0.4 ml of SW480 shluc and SW480 shCA8 cells. Why was the medium collected? These inaccuracies need to be explained.

In the Results section, 3.1. Identification of CA8 and VEGF-A expression in colorectal cancer tissues, the authors present results of IHC analysis of CA VIII, VEGF-A levels in CRC tissue samples Stage I-IV.However, the number of samples for each stage is not specified. What are the levels of CA VIII and VEGF-A in healthy colon tissue?

In Fig1A, information about the magnification used is missing. The presented Stage I results are at a different magnification than the others. How and on what tissue were the antibodies used validated?

Manuscript lacks results 3.4. Section 3.3 is followed by 3.5. The evidence presented in Figures 4 and 5 is not described in the Results section.

In Fig.3, the authors present the results of VEGF mRNA and protein by qPCR and western blot: (B&C) VEGF mRNA and protein expression in CA8 knockdown cell lines (CA8 shRNA) and treatment with miR-16-5p inhibitor by examined by qPCR and western blot . (D&E) VEGF mRNA and protein expression in CA8 overexpression cell lines (CA8 myc) and treatment with miR-16-5p mimic by examined by qPCR and western blot. How did the authors determine the concentration of VEGF in pg /ml from the western blot? Original western blot images should be part of the Original images document.

The authors use non-standard, unclear and inaccurate formulations, e.g. Cells were incubated with different grades of CA8 expression CRC cells for 24 h.

In the Materials and methods section, the authors report the use of 10 nude mice divided into groups of 5. Only 4 mice per group are presented.

Round 2

Reviewer 2 Report

Thank you for answering my questions and explaining the inaccuracies arising from the manuscript v1. In such a form it is possible to accept the manuscript.